# Hidden symmetries, the Bianchi classification and geodesics of the quantum geometric ground-state manifolds

**Diego Liska$^{1\star}$ and Vladimir Gritsev$^{1,2}$**

**1** Institute for Theoretical Physics, Universiteit van Amsterdam,
Science Park 904, Postbus 94485, 1098 XH Amsterdam, The Netherlands
**2** Russian Quantum Center, Skolkovo, Moscow, Russia

$\star$ d.liska@uva.nl

## Abstract

We study the Killing vectors of the quantum ground-state manifold of a parameter-dependent Hamiltonian. We find that the manifold may have symmetries that are not visible at the level of the Hamiltonian and that different quantum phases of matter exhibit different symmetries. We propose a Bianchi-based classification of the various ground-state manifolds using the Lie algebra of the Killing vector fields. Moreover, we explain how to exploit these symmetries to find geodesics and explore their behaviour when crossing critical lines. We briefly discuss the relation between geodesics, energy fluctuations and adiabatic preparation protocols. Our primary example is the anisotropic transverse-field Ising model. We also analyze the Ising limit and find analytic solutions to the geodesic equations for both cases.



# 1 Introduction

In recent years, there has been an increasing interest in the study of the geometry of quantum states of quantum many-body systems. While the origin of the geometric approach for characterising quantum states is rooted in the quantum estimation theory developed in 70's [1, 2], see [3] for a recent review, only relatively recently it became a useful tool for wider applications. Geometric invariants based on quantum geometric tensors have been used to study quantum phase transitions [4–8], to create optimal adiabatic ground-state preparation protocols [9] and to derive bounds for the time integral of energy fluctuations over unit fidelity protocols [10]. The quantum geometric approach became an experimentally testable tool for physics of the many-body ground states and non-equilibrium dynamics in a number of setups [11–18].

The idea behind these works is that quantum mechanics can be viewed as a geometric theory in the following sense. The parameter space of an arbitrary quantum system can be endowed with the structures of Riemannian and differential geometry. The simplest, and most commonly used way, is to introduce a metric in parameter space by considering the overlap amplitude between neighbouring ground states. The resulting object is commonly known as quantum geometric tensor (QGT). The real symmetric part of the QGT, also called quantum Fisher-Rao metric, quantum information metric or, somewhat erroneously Fubini-Study metric, defines a Riemannian metric on the parameter manifold. In contrast, the imaginary part is related to the Berry curvature associated to the Berry connection. Note however that its derivation is entirely generic and does not rely on any adiabatic assumptions. These two complementary parts of the QGT provide a wealth of geometrical and topological structures to study quantum many-body systems. From this metric, we can construct geometric quantities such as Killing vectors, Riemann and Ricci tensors, scalar curvatures, et cetera. Whereas both the real and imaginary parts provide us with topological data of the quantum parameter manifold like the Euler and Chern (or Chern-Simons, depending on dimensionality) invariants. Note that these invariants may abruptly change across phase transitions.

In order to have a better picture of the geometry and the shape of a manifold, it is important to understand its *symmetries*. These are encoded in the so-called Killing vector fields which are intimately related to Lie derivatives. Indeed, these Killing vectors naturally satisfy Lie algebra relations and form the isometry group of the manifold. In 1898 Bianchi (see [19] for a translation of the original text) suggested a classification of low-dimensional (d=1,2,3) Lie algebras which naturally leads to a classification of real and complex manifolds. In 3 dimensions, for example, this distinguishes 11 classes. For later developments and higher dimensions see [20]. In the 80's Thurston conjectured a geometrization program (see the summary book [21]) according to which every closed three-dimensional manifold can be built up out of these Bianchi geometric class model geometries using tools of differential topology. Perelman [22–24] proved the geometrization conjecture in 2003.

Following this line of thought, we arrive at the rather intriguing possibility of a Bianchi-based classification of the parameter manifolds of the *quantum ground states* of many-body systems for (at least) a low number of parameters. As a consequence, *different quantum phases of matter* correspond to different Bianchi classes or can be constructed out of them according to the geometrization conjecture. States corresponding to different classes are separated by quantum

phase transitions. We illustrate this approach here with the example of the quantum transverse-field Ising model (TFIM). This model shows an interesting phenomenon: the *quantum ground state parameter manifold may have symmetries which are not visible at the level of the Hamiltonian*. In particular, one of the phases of the anisotropic TFIM has two continuous symmetry generators while the Hamiltonian itself has only a $\mathbb{Z}_2$ discrete symmetry.

Another facet of the Killing vectors approach is the notion of *geodesics*. For every Killing vector field, there is a quantity which is conserved along geodesics, according to the Nöther theorem. These conserved quantities allow for the explicit integration of the geodesic equations. The latter could aid in the design of optimal quantum state preparation protocols.

Despite the QGT being the "drosophila" of low-dimensional many-body physics, in terms of frequency of study, in both equilibrium and non-equilibrium setups, see e.g. [25] for an extensive review, only a limited number of papers are devoted to the quantum geometric aspects of the QGT, [8–10, 26, 27]. On the other hand, we are not aware of analytical solutions for the geodesic paths of the ground-state manifold of the TFIM spin chain for the full parameter space $(h, \gamma, \phi)$, solutions are only known for two-dimensional sections [28–30]. For this simple integrable model, we can find analytical solutions. In order to solve the geodesic equations, we exploit the symmetries of the manifold. Since the Noether's theorem associates a conserved charge to each symmetry, with enough symmetries, we can constrain the problem completely. Interestingly, we find that *some symmetries are lost during phase transitions*.

The paper is organized as follows: Sections II and III are devoted to covariant formulations of quantum geometric tensors and related geometric quantities, such as geometric tensors, Christoffel symbols, Killing vectors and symmetries; Section IV deals with the transverse field XY model; In Section V we analyze hidden symmetries of the Killing vector fields and the Bianchi classification of the quantum phases, while a special limit of near pure Ising model is treated in Section VI. Geodesics and the energy fluctuations are considered in Section VII. Possible future directions are discussed in Section VIII.

## 2  Geometric tensors

The geometric approach to quantum mechanics sprang from quantum information theory, in the study of quantum parameter estimation [1, 2]. In this setting, a metric, the quantum Fisher information matrix or quantum Fisher-Rao metric, is defined in the space of possibly mixed density matrices $\rho$. This metric is based on the symmetric logarithmic derivative operator formalism. Consider a family of continuous parameters $x^\mu$ such that $\rho = \rho(x)$. The quantum Fisher information matrix is defined as

$$\mathcal{F}_{\mu\nu} = \frac{1}{2} \text{Tr}\big(\rho(x)\{L_\mu, L_\nu\}\big), \tag{1}$$

where $L_\mu$ denotes the symmetric logarithmic derivative whose defining equation is in turn

$$\partial_\mu \rho(x) = \frac{1}{2}\big(\rho L_\mu + L_\mu \rho\big). \tag{2}$$

The Fisher information is equivalent to the Bures metric and it endows the parameter space $x^\mu$ with a Riemannian structure. The statistical distance that this metric defines is related to the quantum fidelity

$$\mathcal{F}_{\mu\nu}dx^\mu dx^\nu = 8\big(1 - \sqrt{F(\rho(x), \rho(x+dx))}\big), \tag{3}$$

where $F(\rho, \sigma) = \big(\text{Tr}\sqrt{\sqrt{\rho}\sigma\sqrt{\rho}}\big)^2$. The Fisher information measures the sensitivity of a quantum state with respect to changes in the parameters $x^\mu$ (assuming one can trace this state through changes in the Hamiltonian, e.g. there is always a GS and a gap).

One of the central results of this theory is that the variance $\text{Var}(x^\mu)$, associated with the estimation of the parameter $x^\mu$ after $M$ independent measurements, satisfies the quantum Cramer-Rao bound

$$\text{Var}(x^\mu) \geq \frac{1}{M\mathcal{F}_{\mu\mu}}. \tag{4}$$

One can consult [3, 25] for a recent review of this topic. The geometrization of quantum mechanics via quantum information theory is robust and has been studied extensively. However, the generality of this approach turns out to be a disadvantage when working with pure states.

Unlike mixed states, the set of pure density matrices, from now on denoted by the projective Hilbert space $P(\mathcal{H})$, is a *Kähler manifold*. In addition to the Riemannian structure coming from the quantum Fisher information, there is a *complex structure* and a *symplectic structure*. To uncover the geometric tensors that define these structures, we take a different route to geometrization and focus our analysis on the properties of the tangent bundle $TP(\mathcal{H})$. We want to emphasize that, in the following, we will only work with pure density matrices $\rho = |\psi\rangle\langle\psi|$. We work with the matrices $\rho(x)$ and not the wavefunctions $|\psi(x)\rangle$ because, as we will see, this will simplify the equations and the results are guaranteed to be gauge invariant. However, as a final step, one can also express all the results in terms of wave functions $|\psi\rangle$. We will discuss some of the subtleties that appear when working with mixed states at the end of this section.

For now, let us assume that the variables $x^\mu$ are a coordinate patch of $P(\mathcal{H})$, i.e. $\dim(x^\mu) = \dim P(\mathcal{H})$. Later, we will restrict the variables $x^\mu$ to a much narrower set of physical parameters. The tangent space $T_\rho P(\mathcal{H})$ at a point $\rho(x)$ is the vector space spanned by the set of matrices

$$t_\mu(x) = \partial_\mu \rho(x). \tag{5}$$

This basis is called the coordinate basis of the tangent bundle. Note that our tangent vectors are Hermitian and traceless matrices. Moreover, if $\rho(x) = |\psi(x)\rangle\langle\psi(x)|$, with $\langle\psi|\psi\rangle = 1$, we have that

$$t_\mu = |\partial_\mu\psi\rangle\langle\psi| + |\psi\rangle\langle\partial_\mu\psi|, \tag{6}$$

where $|\partial_\mu\psi\rangle = \partial_\mu|\psi(x)\rangle$. Since we are working with pure states, $t_\mu = \{\rho, t_\mu\}$, i.e. $t_\mu$ is proportional to the symmetric logarithmic derivative $L_\mu$.

Let us define the linear operator $\mathcal{A}_\mu(x)$ such that $\mathcal{A}_\mu(x)|\psi(x)\rangle = i|\partial_\mu\psi(x)\rangle$. Since $\partial_\mu(\langle\psi|\psi\rangle) = 0$, $\mathcal{A}_\mu(x)$ must be Hermitian, and our tangent vector can be written in terms of $\mathcal{A}_\mu(x)$ as

$$t_\mu = i[\rho(x), \mathcal{A}_\mu(x)], \tag{7}$$

where $[\ ,\ ]$ is the matrix commutator. We conclude that every tangent vector $t_\mu$ is generated by a Hermitian matrix $\mathcal{A}_\mu(x)$. The converse is also true: if $\mathcal{A}(x)$ is a Hermitian matrix, then the commutator $i[\rho(x), \mathcal{A}(x)]$ is a tangent vector. The matrices $\mathcal{A}_\mu(x)$ are called *adiabatic gauge potentials* (AGPs). These potentials are fundamental objects in adiabatic perturbation theory. They also play an essential role in describing the geometry of classical and quantum states. We can even use these potentials to generalize geometric concepts to the case of stationary and non-stationary density matrices. We recommend [31] for a recent review on this topic.

As we saw earlier, the Fisher information matrix defines a metric on the tangent bundle

$$g_{\mu\nu} = \frac{1}{2}\mathcal{F}_{\mu\nu} = \text{Tr}(t_\mu t_\nu). \tag{8}$$

There are multiple equivalent ways to write this equation. In terms of the wave function $|\psi\rangle$,

$$g_{\mu\nu} = 2\,\text{Re}\left[\langle\partial_\mu\psi|\partial_\nu\psi\rangle\right] + 2\langle\partial_\mu\psi|\psi\rangle\langle\partial_\nu\psi|\psi\rangle, \tag{9}$$

and in terms of AGPs the metric reads

$$g_{\mu\nu} = \langle\{\mathcal{A}_\mu, \mathcal{A}_\nu\}\rangle_c = \langle\{\mathcal{A}_\mu, \mathcal{A}_\nu\}\rangle - 2\langle\mathcal{A}_\mu\rangle\langle\mathcal{A}_\nu\rangle, \tag{10}$$

where $\langle X \rangle = \langle \psi | X | \psi \rangle$.

One can check that this formula is gauge invariant. That is, the components of this metric are the same even if we change our basis of kets $|\psi(x)\rangle \to e^{i\phi(x)}|\psi(x)\rangle$. This metric is called *the Fubini-Study metric*. Let us explain the subtle difference between the terms Fisher-Rao metric and Fubini-Study metric. The Fubini-Study metric refers to the Hilbert-Schmidt inner product, or trace product, restricted to the set of pure density matrices. The Fisher-Rao metric, on the other hand, is defined on the set of mixed and pure density matrices via the symmetric logarithmic derivative. The Fisher-Rao metric, when restricted to pure states, reduces to the Fubini-Study metric. Because of this connection, we can relate the Fubini-Study metric to the notion of fidelity susceptibility

$$1 - |\langle \psi(x) | \psi(x+dx)\rangle|^2 = \frac{1}{2} g_{\mu\nu} dx^\mu dx^\nu. \tag{11}$$

This relationship has motivated the study of quantum phase transitions from a geometrical perspective [4, 8, 32].

Recall that an *almost complex structure* in a complex manifold $\mathcal{M}$ is a $(1,1)$-tensor field $J : T_p\mathcal{M} \to \mathcal{M}$ such that $J \circ J = -1$. For $P(\mathcal{H})$, an almost complex structure arises naturally when we consider the vector fields generated by the tangent vectors themselves

$$J(t_\mu) = i[\rho, t_\mu]. \tag{12}$$

Since $t_\mu$ is a Hermitian matrix $J(t_\mu)$ is a tangent vector and $J$ is a well-defined tensor field of rank $(1,1)$. Note that applying the map twice returns the original tangent vector but with the opposite sign

$$J\big(J(t_\mu)\big) = -[\rho, [\rho, t_\mu]] = -t_\mu. \tag{13}$$

This follows from the property $\rho^2 = \rho$ and the relations $\{\rho, t_\mu\} = t_\mu$ and $\rho t_\mu \rho = 0$. Hence, $J$ is an almost complex structure on $P(\mathcal{H})$. This complex structure is compatible with the Fubini-Study metric

$$g(J(t_\mu), J(t_\nu)) = g(t_\mu, t_\nu). \tag{14}$$

A metric that has this property is called a Hermitian metric. Finally, we can use the almost complex structure to define the *symplectic two-form*

$$\Omega(t_\mu, t_\nu) = g(t_\mu, J(t_\nu)) = i \operatorname{Tr}\big([\rho, \partial_\nu \rho]\partial_\mu \rho\big). \tag{15}$$

By using the metric compatibility of $J$ we can show that $\Omega$ is antisymmetric, i.e. it is a differential two-form. Moreover, this two form is non-degenerate because the metric is non-degenerate. If we can prove that $d\Omega = 0$, then we have successfully endowed $P(\mathcal{H})$ with a Kähler structure. Let us first demystify the identity of $\Omega$ by expressing it in terms of the wave function $|\psi(x)\rangle$,

$$\Omega_{\mu\nu} = -i\big(\langle \partial_\nu \psi | \partial_\mu \psi \rangle - \langle \partial_\nu \psi | \partial_\mu \psi \rangle\big), \tag{16}$$

where $\rho(x) = |\psi(x)\rangle\langle\psi(x)|$. This is the Berry curvature, and it is the field strength of the *quantum geometric connection $A_\mu$*

$$A_\mu = i\langle \psi | \partial_\mu \psi \rangle. \tag{17}$$

Note that the quantum geometric connection depends on our choice of phase $e^{i\phi(x)}|\psi(x)\rangle$ (as expected from a gauge field), but the field strength $\Omega = -i dA$ does not. Also observe that $A_\mu$ are the diagonal components of the AGP $\mathcal{A}_\mu$. From this, we also conclude that $d\Omega = 0$, since $d^2 = 0$. This shows that the Fubini-Study metric and the Berry curvature are intimately related. We can express both using a single complex tensor: *the quantum geometric tensor*

$$Q_{\mu\nu} = g_{\mu\nu} + i\Omega_{\mu\nu} = 2\langle \partial_\mu \psi | \partial_\nu \psi \rangle - 2\langle \partial_\mu \psi | \psi \rangle\langle \psi | \partial_\nu \psi \rangle. \tag{18}$$

When working with mixed states, there are a few generalizations that are worth mentioning. We began our discussion on geometry by introducing the Fisher information matrix, a metric defined on the set of mixed states. This metric is equivalent to the Bures metric, and it is related to the quantum fidelity $F(\rho, \sigma)$. However, there are other metrics that we can consider. In dynamical response theory, for example, the definitions that appear naturally are a generalization of the connected correlation functions. For example $\Omega_{\mu\nu} = i \operatorname{Tr}\bigl(\rho_{\text{thermal}}[\mathcal{A}_\mu, \mathcal{A}_\nu]\bigr)$ [31]. These two definitions only coincide when working with pure states and have different properties otherwise. In this paper, we focus on the Riemannian properties of pure states and leave the mixed states' discussion for future work.

## 3 The ground-state manifold

Let us begin our discussion with a Hamiltonian $H(x)$ that depends on a parameter manifold $x^\mu \in \mathcal{M}$. In this section, $\dim(x^\mu) \leq \dim P(\mathcal{H})$. So now, our parameters will only parametrize a submanifold of $P(\mathcal{H})$ and not the entire space. For simplicity, we will assume that our Hamiltonian has a non-degenerate ground state $|\Omega(x)\rangle$. Depending on the specific Hamiltonian, the ground state $|\Omega(x)\rangle$ could be an embedding of $\mathcal{M}$ into $P(\mathcal{H})$ or not. Recall that an embedding is a smooth map that is injective. Sometimes, $|\Omega(x)\rangle$ is independent of a variable $x^\mu$, and therefore the map is not injective. We are interested in the cases in which $|\Omega(x)\rangle$ describes an embedding (at least for a subset $U \subseteq \mathcal{M}$). In other words, we want to study the cases in which the set $\{\rho_0(x) = |\Omega(x)\rangle\langle\Omega(x)| : x \in \mathcal{M}\}$ is a well-defined submanifold of $P(\mathcal{H})$. We call this submanifold the ground-state manifold of $H(x)$. Strictly speaking, the ground-state manifold and the parameter manifold $\mathcal{M}$ are two different spaces but, since we are dealing with an embedding, we will abuse the notation and refer to both as the ground-state manifold $\mathcal{M}$.

What geometric tensors do we have on the ground-state manifold? The pullback of $g$ defines a Riemannian structure on $\mathcal{M}$, but the pullback of $\Omega$ does not always define a symplectic structure. This happens because the pullback of a non-degenerate two-form is not guaranteed to be another non-degenerate two-form. Indeed, if $\mathcal{M}$ is odd dimensional, then $\Omega$ (restricted to the tangent space of the submanifold), is a degenerate two-form. Nonetheless, $\Omega$ still has the interpretation of the Berry curvature. Unfortunately, the pullback of the almost complex structure is not a well defined tensor on $\mathcal{M}$.

We will pay special attention to the Riemannian structure of the ground-state manifold and use this structure to study quantum phase transitions. Given a Riemannian manifold $(\mathcal{M}, g)$ there are a few standard quantities that we can compute: the Riemann tensor and its contractions, Killing vector fields and geodesics. Let us quickly recall the definitions of these objects.

A Killing vector field is the infinitesimal generator of an isometry. From an active point of view, isometries are changes in ground-state manifold that leave the metric invariant. Consider a smooth deformation of our ground-state manifold $\rho(x, \tau)$ driven by the parameter $\tau$ such that $\rho(x, 0) = \rho(x)$. We should think of this deformation as defining a new, deformed, ground-state manifold $\mathcal{M}(\tau) = \{\rho(x, \tau) : x \in \text{parameter space}\}$ for each value of $\tau$. We say that this family of diffeomorphisms is a continuous isometry if

$$\frac{d}{d\tau} g_{\mu\nu}(x, \tau) = \frac{d}{d\tau} \operatorname{Tr}\bigl[\partial_\mu \rho(x, \tau) \partial_\nu \rho(x, \tau)\bigr] = 0. \tag{19}$$

That is, the metric does not change under the transformation. The Killing vector field that generates this isometry is

$$\xi(x) = \frac{d}{d\tau} \rho(x, \tau)\Big|_{\tau=0}. \tag{20}$$

An example of a Killing vector field is the vector field generated by a constant AGP $\mathcal{A}$

$$\xi(x) = i[\rho(x), \mathcal{A}]. \tag{21}$$

We can immediately check this result

$$
\begin{aligned}
\frac{d}{d\tau} g_{\mu\nu}(x, \tau) &= \mathrm{Tr}\left(\partial_\mu \xi(x) t_\nu\right) + \mathrm{Tr}\left(t_\mu \partial_\nu \xi(x)\right) \\
&= \mathrm{Tr}\left([t_\mu, \mathcal{A}] t_\nu\right) + \mathrm{Tr}\left(t_\mu [t_\nu, \mathcal{A}]\right) \\
&= 0.
\end{aligned} \tag{22}
$$

Since we are working with an embedding of $\mathcal{M}$ in an ambient space $P(\mathcal{H})$, we have to consider two types of isometries. If the Killing vector field $\xi \in TP(\mathcal{H})$ is part of the tangent bundle $T\mathcal{M}$, i.e. $\xi = \xi^\mu t_\mu$ for a coordinate basis $\{t_\mu = \partial_\mu \rho\}$ of $T\mathcal{M}$ then the submanifold is invariant under the isometry and $\xi$ satisfies the Killing equation

$$\mathcal{L}_\xi g_{\mu\nu} = \nabla_\mu \xi_\nu + \nabla_\nu \xi_\mu = 0. \tag{23}$$

Where $\mathcal{L}$ denotes the Lie derivative. If $\xi \in TP(\mathcal{H})$ but not in $T\mathcal{M}$ then the isometry does not leave the submanifold invariant. You may think of a rotation that leaves the 2-sphere embedded in $\mathbb{R}^3$ invariant and a translation that changes its position in space. Both are isometries but the Killing vector field of the rotation lies inside the tangent bundle of the 2-sphere and the Killing vector field of the translation does not. We are mostly concerned with the first class of isometries thus, will also require the Killing vector field to be part of the tangent bundle of $\mathcal{M}$.

The Killing equation can also be written in terms of wave functions $|\psi\rangle$ or in terms of AGPs $\mathcal{A}_\mu$. For example a vector field $\xi = i[\rho(x), \mathcal{A}_\xi(x)]$ is a Killing vector field if and only if

$$\langle\{\partial_\mu \mathcal{A}_\xi, \mathcal{A}_\nu\}\rangle_c + \langle\{\partial_\nu \mathcal{A}_\xi, \mathcal{A}_\mu\}\rangle_c = 0. \tag{24}$$

The set of Killing vectors on a manifold $\mathcal{M}$ forms a Lie algebra, whose Lie bracket is defined by the differential commutator $[\![\ ,\ ]\!]$. This commutator should not be confused with the matrix commutator $[\ ,\ ]$. If $t_1 = t_1^\mu(x) \partial_\mu \rho(x)$ and $t_2(x) = t_2^\mu(x) \partial_\mu \rho(x)$ are two vector fields on $\mathcal{M}$, then

$$[\![t_1, t_2]\!] f(x) = t_1^\mu \partial_\mu \left[t_2^\nu \partial_\nu f(x)\right] - t_2^\mu \partial_\mu \left[t_1^\nu \partial_\nu f(x)\right], \tag{25}$$

where $f : \mathcal{M} \to \mathbb{R}$ is a test function on $\mathcal{M}$. Low dimensional, real Lie algebras (d = 1, 2, 3) have been classified. In 3 dimensions, for example, there are 11 classes. This is called the Bianchi classification [19]. This suggest the possibility of classifying the different quantum ground-states of many-body systems using the Lie algebra of their Killing vector fields.

Geodesics are paths that locally minimize the distance between two points in a manifold. We can find them by solving the geodesic equations

$$\frac{d^2 x^\mu}{ds^2} = -\Gamma^\mu_{\lambda\rho} \frac{dx^\lambda}{ds} \frac{dx^\rho}{ds}. \tag{26}$$

Here $s$ is an affine parameter, i.e. $g_{\mu\nu} \frac{dx^\mu}{ds} \frac{dx^\mu}{ds} = 1$. Most of the time, we can only solve these equations numerically. One exception happens when we have enough Killing vector fields in our manifold. Each Killing vector has an associated conserved charge along geodesics $x^\mu(s)$

$$\partial_s Q_\xi = \partial_s \left(\xi_\mu \frac{dx^\mu}{ds}\right) = 0. \tag{27}$$

So, each Killing vector corresponds to a first order differential equation. Requiring that our geodesic is parametrized by an affine parameter gives one extra restriction. In general, we only need $\dim(\mathcal{M}) - 1$ Killing vector fields to find the geodesics of a manifold.

For completeness, let us recall the equations for the Christoffel symbols and the Riemann tensor. We will not directly discuss these quantities in this paper, but they are important and useful concepts in quantum geometry. Recall that the Christoffel symbols, are given by the formula

$$\Gamma^\lambda_{\mu\nu} = \frac{1}{2} g^{\lambda\delta} (\partial_\nu g_{\mu\delta} + \partial_\mu g_{\nu\delta} - \partial_\delta g_{\mu\nu}). \tag{28}$$

We can take an advantage that we are working with an embedding and write an expression for these symbols in terms of traces and tangent vectors:

$$\Gamma_{\lambda\mu\nu} = g_{\lambda\delta} \Gamma^\delta_{\mu\nu} = \mathrm{Tr}\left(t_\lambda \partial_\mu t_\nu\right). \tag{29}$$

We show how to derive this expression in Appendix A.

The Riemann tensor and its contractions encode all the information about the curvature of the manifold. In a coordinate basis, the components of the Riemann tensor are given by

$$R^\rho_{\sigma\mu\nu} = \partial_\mu \Gamma^\rho_{\nu\sigma} + \Gamma^\rho_{\mu\lambda} \Gamma^\lambda_{\nu\sigma} - (\mu \leftrightarrow \nu). \tag{30}$$

The contractions of the Riemann tensor are commonly known as the Ricci tensor $R_{\mu\nu} = R^\lambda_{\mu\lambda\nu}$ and the Ricci scalar $R = R^\mu_\mu$. The Riemann tensor also has topological information, since its integral gives the Euler characteristic of the manifold. This result is known as the Gauss-Bonnet theorem in two dimensions [33] and the Chern-Gauss-Bonnet theorem in any number of even dimensions [34]. In [8], the authors proposed to use the Euler characteristic of the ground-state manifold as a new topological number.

# 4 The anisotropic transverse-field Ising model

Let us apply these concepts to the anisotropic TFIM, also known as the XY model. There are a few reasons why this model is a good example. First, we can solve the model exactly. Second, this model has a rich phase diagram with three different regions: two ferromagnetic phases and one paramagnetic phase. Third, the Hamiltonian depends on three parameters and has a non-vanishing Berry curvature. The model is described by the Hamiltonian

$$H = -\sum_{j=1}^{N} J_x \sigma^x_j \sigma^x_{j+1} + J_y \sigma^y_j \sigma^y_{j+1} + h\sigma^z_j, \tag{31}$$

where $\sigma^\alpha_j$ are the Pauli matrices of the $j$-th spin site. To fix our energy scale we work with the variables:

$$J_x = J\left(\frac{1+\gamma}{2}\right) \text{ and } J_y = J\left(\frac{1-\gamma}{2}\right), \tag{32}$$

and set $J = 1$. We will add an additional parameter $\phi$ to our Hamiltonian that corresponds to a rotation of all spins around the z-axis by an angle of $\phi/2$. We apply this rotation with the unitary transformation $U = \prod_j e^{-i\phi\sigma^z_j/4}$, $H \to UHU^\dagger$. We assume periodic boundary conditions, $\sigma^\alpha_{N+1} = \sigma^\alpha_1$. The solution of this model is somewhat convoluted and it involves Jordan-Wigner, Fourier and Bogoliubov transformations. We will not solve the model here but the interested reader may find a modern version of the solution in [27]. The mapping to fermions yields a unique ground state that can be represented using a tensor product of Bloch vectors with polar angle $\theta$ and azimuthal angle $\phi$:

$$|\Omega(h,\gamma,\phi)\rangle = \prod_{k>0} |\Omega_k\rangle \quad \text{with} \quad |\Omega_k\rangle = \begin{pmatrix} \cos(\theta_k/2)e^{-i\frac{\phi}{2}} \\ \sin(\theta_k/2)e^{i\frac{\phi}{2}} \end{pmatrix}. \tag{33}$$

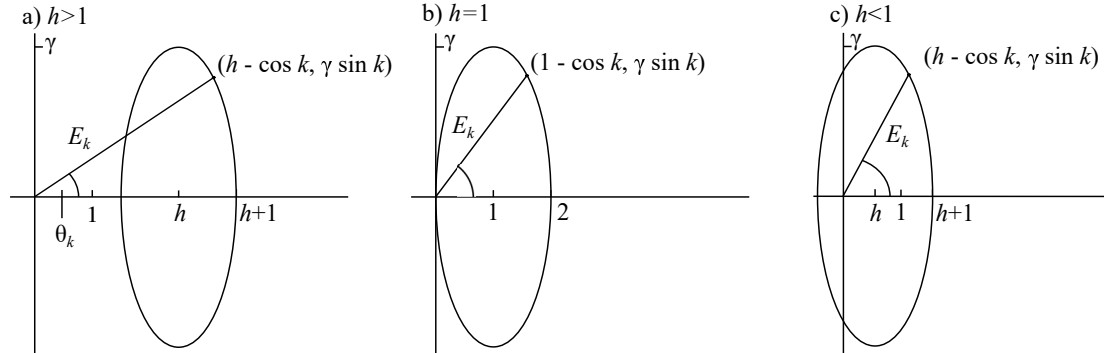

Figure 1: Ellipse representation of the anisotropic TFIM ground state. Note that the ellipse is parametrized counterclockwise whenever $\gamma > 0$. The winding number $|\theta_\pi - \theta_0|/\pi$ determines if the Hamiltonian is in the paramagnetic or ferromagnetic phase.

Here,

$$E_k = \sqrt{(h - \cos k)^2 + \gamma^2 \sin^2 k} \quad \text{and} \quad \tan \theta_k = \frac{\gamma \sin k}{h - \cos k}. \tag{34}$$

Your may see [8] for a detailed description of this ground-state. The Brillouin zone of this system is $k = 2\pi n/N$, $n = 1, 2, \ldots, N$ or equivalently $k = 2\pi n/N - \pi$. For simplicity, we assume $N$ is an even number, so that every $k > 0$ is given by $k = \pi n/(N/2)$, for $n = 1, 2, \ldots, N/2$. These definitions suggest a graphic representation for the ground state $|\Omega(h, \gamma, \phi)\rangle$ as a loop in the $xy$-plane. If we interpret the energy $E_k$ as a distance from the origin of the $xy$-plane and $\theta_k$ as its angle from the x-axis, we find that the allowed energies lie on the ellipse

$$x(k) = h - \cos k \quad \text{and} \quad y(k) = \gamma \sin k. \tag{35}$$

The allowed energies depend on the values of $h$ and $\gamma$. There are a few combinations of $h$ and $\gamma$ that are important (see Fig. 1). When $h = 1$ the ellipse touches the origin. At this point, our model is a gapless theory. The critical line $h = 1$ separates the ferromagnetic ($h < 1$) and paramagnetic ($h > 1$) phases. In the paramagnetic region, the ellipse also touches the origin when $\gamma = 0$. This is an example of an anisotropic phase transition between a ferromagnet aligned along the X direction ($\gamma > 1$) and a ferromagnet aligned in the Y direction ($\gamma < 1$).

We can associate a topological number to the ferromagnetic and paramagnetic phases: the winding number of the ellipse $(x(k), y(k))$ with respect to the origin. A winding number of 1 indicates a ferromagnetic ground state, while a winding number of 0 indicates a paramagnetic one. An analysis of these shapes and their topological properties can be found in [35]. These shapes are widely used in the study of extended TFIM, see, for example, [36].

Now that we have a ground-state manifold, let us compute its metric. A few properties of $|\Omega_k\rangle$ simplify the computation: first $\langle \partial_\mu \Omega_k | \Omega_k \rangle = 0$ and second $\text{Re}[\langle \partial_\phi \Omega_k | \partial_\mu \Omega_k \rangle] = 0$. Where $\mu, \nu = h, \gamma$. We find that the components of the metric are

$$g_{\mu\nu} = \frac{1}{2} \sum_{k>0} (\partial_\mu \theta_k)(\partial_\nu \theta_k), \tag{36}$$

and for $\phi$, we have

$$g_{\phi\mu} = 0, \quad g_{\phi\phi} = \frac{1}{2} \sum_{k>0} 1 - \cos^2 \theta_k = \frac{1}{2} \sum \sin^2 \theta_k. \tag{37}$$

Using Fig. 1, it is easy to find explicit expressions for the metric

$$g_{h\gamma} = \sum_{k>0} \frac{\gamma(\cos k - h)\sin^2 k}{2E_k^4}, \qquad g_{hh} = \sum_{k>0} \frac{\gamma^2 \sin^2 k}{2E_k^4},$$

$$g_{\gamma\gamma} = \sum_{k>0} \frac{(h - \cos k)^2 \sin^2 k}{2E_k^4}, \qquad g_{\phi\phi} = \sum_{k>0} \frac{\gamma^2 \sin^2 k}{2E_k^2}. \tag{38}$$

The corresponding expressions for the Berry curvature are

$$\Omega_{\gamma\phi} = \sum_{k>0} \frac{\gamma(\cos k - h)\sin^2 k}{2E_k^3}, \quad \Omega_{h\phi} = \sum_{k>0} \frac{\gamma^2 \sin^2 k}{2E_k^3}. \tag{39}$$

These expressions may be evaluated by solving six integrals in the thermodynamic limit, and results have been widely studied in recent years, e.g. [4, 8, 27, 31]. However, in this paper, let us note that not all sums are independent and it turns out we only need to evaluate three of them. The following results are valid in the thermodynamic limit ($N \to \infty$):

$$S_1 = \frac{1}{N} \sum_{n=1}^{N/2} \frac{\gamma^2 \sin^2\left(\frac{2n\pi}{N}\right)}{E\left(\frac{2n\pi}{N}\right)^2} = \frac{1}{2} \begin{cases} \frac{|\gamma|}{1+|\gamma|} & |h| < 1 \\ \frac{\gamma^2}{\gamma^2 - 1}\left(1 - \frac{|h|}{\sqrt{h^2 + \gamma^2 - 1}}\right) & |h| > 1 \end{cases}, \tag{40}$$

where $E(k) = E_k$.

$$S_2 = \frac{1}{N} \sum_{n=1}^{N/2} \frac{1}{E\left(\frac{2n\pi}{N}\right)^2} = \frac{1}{2} \begin{cases} \frac{1}{|\gamma|(1-h^2)} & |h| < 1 \\ \frac{|h|}{(h^2-1)\sqrt{h^2+\gamma^2-1}} & |h| > 1 \end{cases}. \tag{41}$$

The last sum is a complicated expression. It corresponds to the ground-state energy of the model and we need it to compute the components of the Berry curvature.

$$S_3 = \frac{1}{N} \sum_{n=1}^{N/2} E\left(\frac{2n\pi}{N}\right) = \begin{cases} \frac{\sqrt{1-h^2}}{\pi}\left[E(a) - K(a) + \frac{1}{1-h^2}\Pi\left(\frac{h^2}{h^2-1}, a\right)\right], & h^2 + \gamma^2 < 1 \\ \frac{1-h^2}{\pi|\gamma|}\left[\frac{\gamma^2}{1-h^2}E(b) - K(b) + \Pi(h^2, b)\right], & h^2 + \gamma^2 > 1, |h| < 1 \\ \frac{h^2-1}{\pi\sqrt{h^2+\gamma^2-1}}\left[aE\left(\frac{1}{b}\right) - K\left(\frac{1}{b}\right) + \Pi\left(\frac{1}{h^2}, \frac{1}{b}\right)\right], & |h| > 1 \end{cases}, \tag{42}$$

where $E, K$ and $\Pi$ are the elliptic integrals

$$K(a) = \int_0^{\pi/2} \frac{d\theta}{\sqrt{1 - a\sin^2\theta}}, \quad E(a) = \int_0^{\pi/2} d\theta \sqrt{1 - a\sin^2\theta},$$

$$\Pi(n, a) = \int_0^{\frac{\pi}{2}} \frac{d\theta}{(1 - n\sin^2\theta)\sqrt{1 - a\sin^2\theta}} \tag{43}$$

and

$$a = \frac{1 - h^2 - \gamma^2}{1 - h^2}, \qquad b = \frac{h^2 + \gamma^2 - 1}{\gamma^2}. \tag{44}$$

The authors of [37] found this expression and showed that, despite its appearance, it is a smooth function at the line $h^2 + \gamma^2 = 1$.

Let us evaluate the components of the metric tensor. We will divide the components of the metric by the system size $g \to g/N$ and take the thermodynamic limit $N \to \infty$. We find that

$$g_{\phi\phi} = \frac{1}{2}S_1 = \frac{1}{4} \begin{cases} \frac{|\gamma|}{1+|\gamma|} & |h| < 1 \\ \frac{\gamma^2}{\gamma^2-1}\left(1 - \frac{|h|}{\sqrt{h^2+\gamma^2-1}}\right) & |h| > 1 \end{cases}$$

$$g_{hh} = -\frac{\gamma}{4}\frac{\partial S_2}{\partial \gamma} = \frac{1}{8}\begin{cases} \dfrac{1}{|\gamma|(1-h^2)} & |h|<1 \\[2ex] \dfrac{\gamma^2|h|}{(h^2-1)(\gamma^2+h^2-1)^{3/2}} & |h|>1 \end{cases}$$

$$g_{h\gamma} = \frac{1}{4\gamma}\frac{\partial S_1}{\partial h} = \frac{1}{8}\begin{cases} 0 & |h|<1 \\[2ex] -\dfrac{h\gamma}{|h|(\gamma^2+h^2-1)^{3/2}} & |h|>1 \end{cases}$$

$$g_{\gamma\gamma} = \frac{1}{4\gamma}\frac{\partial S_1}{\partial \gamma} = \frac{1}{8}\begin{cases} \dfrac{1}{|\gamma|(|\gamma|+1)^2} & |h|<1 \\[2ex] \left[\dfrac{2}{(1-\gamma^2)^2}\left(\dfrac{|h|}{\sqrt{\gamma^2+h^2-1}}-1\right)\right. \\ \left.-\dfrac{\gamma^2|h|}{(1-\gamma^2)(\gamma^2+h^2-1)^{3/2}}\right] & |h|>1 \end{cases} \tag{45}$$

We will focus on the Riemannian structure defined by $g$, so we do not evaluate the components of the Berry curvature explicitly. However, these can be derived from $S_3$:

$$\Omega_{h\phi} = \partial_h^2 S_3, \qquad \Omega_{\gamma\phi} = \partial_h \partial_\gamma S_3. \tag{46}$$

The components of the metric tensor we present here, and the ones derived in other papers, e.g. [8], differ by a factor of 2. This depends on the convention used for the metric tensor. We work with the trace product whilst many authors prefer to work with half of the trace product.

## 5   Hidden symmetries and Killing vector fields

Let us focus momentarily on the ferromagnetic sector of the ground-state manifold ($|h| < 1$). A simple coordinate transformation reveals a hidden symmetry in the model. Take

$$h \rightarrow \sin u \quad \text{and} \quad \gamma \rightarrow \text{sgn}(v)\tan^2 v, \tag{47}$$

where $u, v \in (-\pi/2, \pi/2)$. The transformed metric reads

$$ds_{\text{ferr}}^2 = \frac{1}{8}(4dv^2 + \cot^2 v\, du^2 + 2\sin^2 v\, d\phi^2). \tag{48}$$

Remarkably, this metric is independent of the variable $u$, meaning that

$$\frac{\partial}{\partial u} = \sqrt{1-h^2}\frac{\partial}{\partial h}, \tag{49}$$

is a Killing vector field on the paramagnetic sector of the ground-state manifold. Note that the vector field $\partial_\phi$ is also a Killing vector field on this sector, and in fact of the entire ground-state manifold. This is no surprise, and the reason is quite simple:

$$|\Omega(h,\gamma,\phi)\rangle = \prod_j e^{-i\phi\sigma_j^z/4}|\Omega(h,\gamma,0)\rangle, \tag{50}$$

where $U = \prod_j e^{-i\phi\sigma_j^z/4}$ is the unitary operator generated by $\mathcal{A}_\phi = \sum_j \sigma_j^z$. The generator of the transformation is also the generator of the vector field $\partial_\phi \rho(x)$, i.e.

$$\partial_\phi \rho = i[\rho, \mathcal{A}_\phi]. \tag{51}$$

Since $\mathcal{A}_\phi(x) = \mathcal{A}_\phi$ is a constant AGP, we conclude that $\partial_\phi \rho$ must be a Killing vector field on the ground-state manifold (see Eq. 22).

Eq. 49 is perhaps the most important result of this paper. It is striking that this Killing vector field exists, since the transformation $u \to u + a$, for $a$ constant, is not a symmetry of the Hamiltonian. It changes the ground-state and its energy. Moreover, the Killing vector is confined to the ferromagnetic part of the ground-state manifold, and the symmetry is lost once we cross the critical line at $h = 1$.

Since both Killing vector fields $\partial_u$ and $\partial_\phi$ correspond to partial derivatives, their Lie algebra

$$[[\partial_\phi, \partial_u]] = 0, \tag{52}$$

corresponds to the Lie algebra of the abelian group $(\mathbb{R}^2, +)$. Here, $[[\ ,\ ]]$ is the commutator of differential operators, not to be confused with the matrix commutator $[,]$. The paramagnetic region of the ground state has only one Killing vector field $\partial_\phi$, and therefore, is isomorphic to the abelian algebra of the group $(\mathbb{R}, +)$. The fact that we can associate a Lie algebra to the different phases of matter suggest the possibility of a Bianchi-based classification of the different quantum phases of matter.

## 5.1 Critical lines and RG flows

Near the Ising phase transition at $|h| = 1$, the low energy TFIM is effectively described by a theory of Majorana fermions whose mass gap is proportional to $|h - 1|$. The arguments of Venuti and Zanardi [5] imply that $g_{hh} \sim |h-1|^{-1}$ whilst $g_{\mu\nu} \sim 1$ for the rest of the components. This argument is based on a simple scaling analysis on the operators associated with the deformations of $x^\mu$. More elaborate arguments, such as the ones presented in [38], give a relationship between Renomalization Group flows, homothetic vector fields and the scaling properties of the quantum metric tensor. However, we argue that this information alone is not enough to determine the Killing vector fields of the ground-state manifold. We can immediately see this from the exact expression for the metric tensor. Close to the critical line, all the components of the metric tensor coincide except the cross term $g_{h\gamma}$. This term is zero in the ferromagnetic manifold and is not zero (and also not divergent) in the paramagnetic manifold. This change alone is enough to spoil the symmetry and prevents the vector field $\partial_u$ to be a Killing vector field, even approximately, in the paramagnetic manifold. That is, knowing that $g_{\mu\nu} \sim 1$ is not enough information to fix the isometries of the manifold.

## 5.2 Geodesics

We have two Killing vector fields in the ferromagnetic manifold. Each associated with a conserved charge along geodesics. Together with the arc-length parametrization condition, we have three first-order differential equations

$$\mathcal{Q}_u = \frac{1}{8} \cot^2 \nu u'(s), \qquad \mathcal{Q}_\phi = \frac{1}{4} \sin^2 \nu \phi'(s) \tag{53}$$

$$\frac{1}{2} v'(s)^2 + \frac{1}{8} \cot^2 \nu\, u'(s)^2 + \frac{1}{4} \sin^2 \nu\ \phi'(s)^2 = 1,$$

that suffice to solve for the geodesics of the manifold. Note that we do not have to consider the geodesic equations, the symmetries give us enough constrains. In terms of the physical variables $h$ and $\gamma$, the conserved charges read

$$\frac{h'(s)}{8|\gamma|\sqrt{1-h^2}} = \mathcal{Q}_u \quad \text{and} \quad \frac{|\gamma|}{4(1+|\gamma|)}\phi'(s) = \mathcal{Q}_\phi. \tag{54}$$

These conserved quantities have also been found in [28] and in [29] by using Euler-Lagrange equations of motion. To understand these equation better, we need an input from numerical

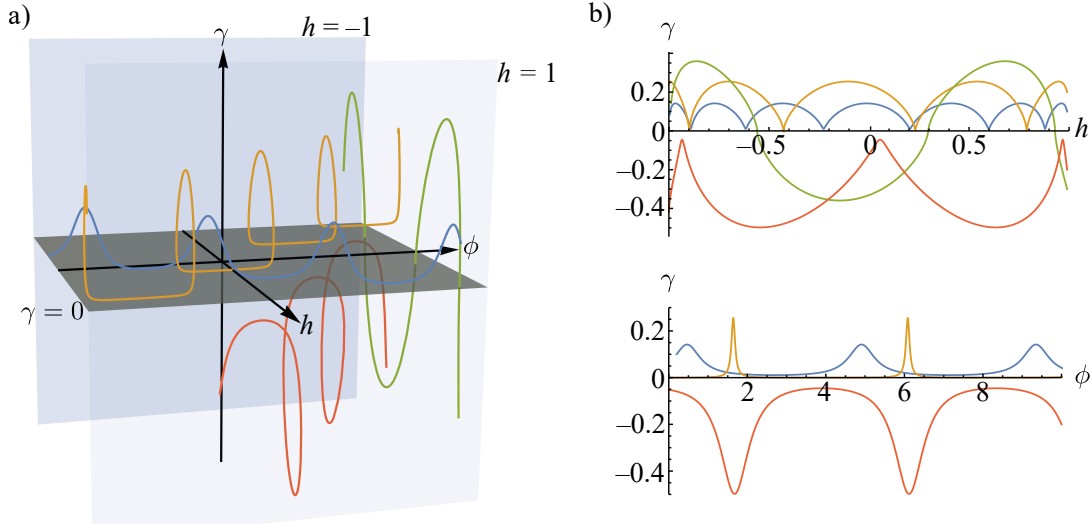

Figure 2: a) Geodesics in the ferromagnetic sector of the ground-state manifold. The different solutions have $\mathcal{Q}_u = -0.9, -0.7, -0.6, -0.47$ and $\mathcal{Q}_\phi = 0.05, -0.01, 0, -0.1$ for blue, orange, green and red respectively. b) The projections onto the $uv$-plane and $\phi u$-plane. Note that only the green curve with $\mathcal{Q}_\phi = 0$ and constant $\phi(s)$ probes the two ferromagnetic phases. All solutions eventually touch the $h = \pm 1$ planes.

solutions. Fig. 2 shows four geodesic solutions. Note that solutions are generically confined to one of the ferromagnetic regions of the ground-state manifold unless there is a fine tuning involved, $\mathcal{Q}_\phi = 0$. When $\mathcal{Q}_\phi \neq 0$, $\gamma(s) > \epsilon$ for a positive $\epsilon$. We prove this statement below. Hence, solutions do not usually touch or cross the critical line at $v = 0$ or $\gamma = 0$. However, they always touch, and cross, the critical lines at $u = \pm\pi/2$ or $|h| = 1$.

If we restrict ourselves to the domain where the functions $u(v)$ and $\phi(v)$ are well defined, solving for $u'(v)$ and $\phi'(v)$, we find that

$$\frac{du}{dv} = \pm \frac{8\mathcal{Q}_u \tan^2 v}{\sqrt{2 - 8\mathcal{Q}_\phi^2 \csc^2 v - 16\mathcal{Q}_u^2 \tan^2 v}} \tag{55}$$

$$\frac{d\phi}{dv} = \pm \frac{4\mathcal{Q}_\phi \csc^2 v}{\sqrt{2 - 8\mathcal{Q}_\phi^2 \csc^2 v - 16\mathcal{Q}_u^2 \tan^2 v}}. \tag{56}$$

We are interested in studying the behaviour of geodesics that cross the phase transition at $v = 0$. The values of $v$ where the derivative diverge correspond to the maximum and minimum values of $v$ a geodesic has. Note that $\csc v \to \infty$ when $v \to 0$, so a geodesic crossing the critical line $v = 0$ must have $\mathcal{Q}_\phi = 0$. Fig. 2 shows an example of a solution with $\mathcal{Q}_\phi = 0$.

A few references [28–30] have suggested that the geodesics inside the $h\gamma$-plane, i.e. solutions with $\mathcal{Q}_\phi = 0$, do not cross the critical line $\gamma = 0$ but only touch it. However, if we have two geodesics, one with $\gamma_1(s) \geq 0$ and the other one with $\gamma_2(s) \leq 0$, we can connect them as long as $h_1(\gamma = 0) = h_2(\gamma = 0)$. The extension of a geodesic inside the upper plane is uniquely specified by demanding the lower plane geodesic to have the same conserved charge $\mathcal{Q}_u$. That is, geodesics with $\mathcal{Q}_\phi = 0$ do cross the $\gamma = 0$ critical line and connect the two ferromagnetic phases.

Interestingly, at $v = 0$ the derivative $u'(v)$ vanishes independently of the value of $\mathcal{Q}_u$. This means that, at the critical line, a geodesic is not uniquely specified by its position and its velocity and we need to take into account higher derivatives. We can explicitly see this behaviour by

doing a Taylor expansion of the geodesic path solution around $v = 0$,

$$u(v) = u(0) + \frac{4\sqrt{2}\mathcal{Q}_u}{3}v^3 + \mathcal{O}\left(v^5\right). \tag{57}$$

Recall that $u = \arcsin(h)$ and $v = \text{sgn}(\gamma)\sqrt{|\gamma|} + \mathcal{O}\left(|\gamma|^{\frac{3}{2}}\right)$. The intuition behind this behavior is quite simple. Near the phase transition, the distance between two points is $8\Delta s^2 \approx (\Delta v)^2 + \cot^2 v (\Delta u)^2$. Since $\cot^2 v \to \infty$ when $v \to 0$, $\Delta u$ must go to zero if we want to have a small value of $s$ after crossing the critical line.

## 6 Near the Ising limit

A detailed analysis of the paramagnetic ground-state manifold is challenging due to the complexity of the metric. Part of the complexity lies in the non-vanishing cross term $g_{\gamma h}$. Due to this term the Killing vector field $\partial_u$ is lost during the phase transition. Even the conserved charge $\mathcal{Q}_\phi = g_{\phi\phi}\phi'(s)$ has a complicated structure. To simplify the metric, we will restrict ourselves to the parameters $(h, \phi)$ and work with a constant value of $\gamma$ (i.e. $d\gamma = 0$). We will refer to this manifold as the $h\phi$-ground-state manifold. We will work near the Ising limit $\gamma \to 1$.

First, let us do the coordinate transformation $h \to \csc\psi$ to clean the metric in the paramagnetic manifold $|h| > 1$. Here, $\psi \in (0, \pi/2)$. The resulting metric is

$$ds^2_{\text{par}} = \frac{1}{8}(d\psi^2 + \sin^2\psi d\phi^2) + \frac{1}{16}\left[\frac{1}{2}(5 + 3\cos 2\psi)\sin^2\psi d\phi^2 \right. \tag{58}$$

$$\left. + (1 + 3\cos 2\psi)d\psi^2\right](\gamma - 1) + \mathcal{O}\left[(\gamma - 1)^2\right].$$

Although it looks messy, another change of variables $\psi \to \beta - \frac{3}{8}(\gamma - 1)\sin 2\beta$, for $\beta \in (0, \pi/2)$, reveals that this is the metric of a 2-sphere

$$ds^2_{\text{par}} = \frac{1+\gamma}{16}(d\beta^2 + \sin^2\beta d\phi^2) + \mathcal{O}\left[(\gamma - 1)^2\right]. \tag{59}$$

The full coordinate transformation reads $h \to \csc\beta + \frac{3}{4}(\gamma - 1)\cos\beta\cot\beta$, for $h > 1$. The 2-sphere is a maximally symmetric space with three Killing vector fields:

$$\xi_1 = \cos\phi\partial_\beta - \cot\beta\sin\phi\partial_\phi, \quad \xi_2 = -\sin\phi\partial_\beta - \cot\beta\cos\phi\partial_\phi, \quad \xi_3 = \partial_\phi. \tag{60}$$

The Lie algebra of these Killing vector fields is the familiar algebra $\mathfrak{so}(3)$ algebra

$$[\![\xi_i, \xi_j]\!] = \varepsilon_{ijk}\xi_k, \tag{61}$$

which corresponds to a Type IX Lie algebra according to the Bianchi classification.

On the other hand, the ferromagnetic part of the $h\phi$-ground-state manifold is a cylinder

$$ds^2_{\text{ferr}} = \frac{1}{8}\left(\frac{du^2}{\gamma} + \frac{2\gamma d\phi^2}{1+\gamma}\right). \tag{62}$$

Here $u = \arcsin(h)$ and $\gamma$ is constant but not necessarily close to one. Again, this is a maximally symmetric space with three Killing vector fields

$$\chi_1 = \sqrt{\gamma}\partial_u, \quad \chi_2 = \sqrt{\frac{1+\gamma}{2\gamma}}\partial_\phi, \quad \chi_3 = -\sqrt{\frac{2\gamma^2}{1+\gamma}}\partial_u + \sqrt{\frac{1+\gamma}{2\gamma^2}}\partial_\phi. \tag{63}$$

The Lie algebra of these Killing vectors is the algebra of the isometries of the Euclidean plane $\mathfrak{e}(2)$:

$$[[\chi_1, \chi_2]] = 0, \quad [[\chi_1, \chi_3]] = \chi_2, \quad [[\chi_2, \chi_3]] = -\chi_1. \tag{64}$$

In the Bianchi classification, this is a Type VII$_0$ Lie algebra.

Note that, despite having restricted ourselves to a hyperplane of the original ground-state manifold of the anisotropic TFIM, we still find that different quantum phases of matter correspond to different algebras.

## 6.1 Geodesics

Near the Ising point $\gamma \approx 1$ the metric of the $h\phi$-ground-state manifold is that of a cylinder for $|h| < 1$ and a 2-sphere for $|h| > 1$. So, in the ferromagnetic manifold, geodesics are linear functions of the type $\phi = mu + b$, for some constants $a$ and $b$. In the paramagnetic manifold, geodesics are great circles, characterized by the implicit equation $\cot \beta = q \cos(\phi + \phi_o)$, for some other constants $q, \phi_o \in \mathbb{R}$. The matching conditions at the boundary give a relationship between the two constants.

$$\cos\left[\phi_o - \phi\left(u = \pm\frac{\pi}{2}\right)\right] = 0, \quad q = -\frac{7 - 3\gamma_o}{4m}. \tag{65}$$

These conditions guarantee that geodesics are differentiable functions with a continuous first derivative.

We can visualize the geodesics of the $h\phi$-ground-state manifold using an isometric embedding of the plane $(h, \phi)$ into $\mathbb{R}^3$. This technique has been used to visualize the topological properties of the TFIM [8] and it is also useful to visualize geodesics. Taking advantage of the rotational symmetry we parametrize our manifold as a surface of revolution

$$\Phi(h, \phi) = \big(g(h), f(h)\cos\phi, f(h)\sin\phi\big). \tag{66}$$

Our task now is to find the functions $f(h)$ and $g(h)$ such that the induced metric

$$
\begin{aligned}
g_{\phi\phi} &= \partial_\phi \Phi \cdot \partial_\phi \Phi = f(h)^2 \\
g_{hh} &= \partial_h \Phi \cdot \partial_h \Phi = f'(h)^2 + g'(h)^2 \\
g_{h\phi} &= \partial_\phi \Phi \cdot \partial_h \Phi = 0,
\end{aligned}
\tag{67}
$$

corresponds to our metric. We find the following system of differential equations for $h > 1$

$$f(h)^2 = \frac{1}{8h^2} + (\gamma - 1)\frac{4h^2 - 3}{16h^4} + \mathcal{O}\big[(\gamma - 1)^2\big] \tag{68}$$

and

$$f'(h)^2 + g'(h)^2 = \frac{1}{8h^2(h^2 - 1)} + (\gamma - 1)\frac{2h^2 - 1}{8h^4(h^2 - 1)} + \mathcal{O}\big[(\gamma - 1)^2\big]. \tag{69}$$

Note that having a surface of revolution simplifies the computation and gives us a direct result for $f(h)$. For $h < 1$ we have the set of equations

$$f(h)^2 = \frac{\gamma}{4(\gamma + 1)}, \quad f'(h)^2 + g'(h)^2 = \frac{1}{8\gamma(1 - h^2)}. \tag{70}$$

Continuity in $f(h)$ requires that

$$\frac{1 + \gamma}{16} = \frac{\gamma}{4(\gamma + 1)}. \tag{71}$$

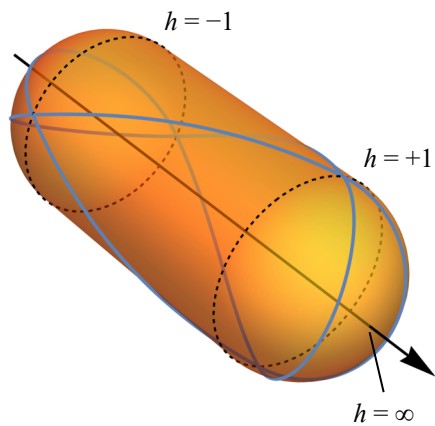

Figure 3: Isometric embedding of the $h\phi$-ground-state manifold with a geodesic path.

This condition can only be fulfilled if $\gamma = 1$. For other values of $\gamma$ an isometric and continuous embedding into $\mathbb{R}^3$ does not exist (at least as a surface of revolution). Solving the differential equations for $\gamma = 1$ we find that

$$f(h) = \frac{1}{\sqrt{2}} \begin{cases} 1 & |h| < 1 \\ |h|^{-1} & |h| > 1 \end{cases} \tag{72}$$

$$g(h) = -\frac{1}{2\sqrt{2}} \begin{cases} \arcsin h & |h| < 1 \\ \dfrac{\sqrt{h^2 - 1}}{h} + \mathrm{sgn}(h)\dfrac{\pi}{2} & |h| > 1 \end{cases}. \tag{73}$$

The embedding corresponds to a cigar-like surface made from a cylinder with two spherical caps. See Fig. 3.

## 7 Geodesics and energy fluctuations

The ideas presented in this paper appear to be somewhat abstract, however they are very physical. For example, we can apply these concepts to develop better ground-state preparation protocols [9, 10, 39]. Consider a parameter-dependent Hamiltonian $H(x^\mu)$ and imagine that we have a system in the ground state $\left|\Omega(x_i^\mu)\right\rangle$ of $H(x_i^\mu)$. We can change the system's state from one ground state $\left|\Omega(x_i^\mu)\right\rangle$ to another one $\left|\Omega(x_f^\mu)\right\rangle$ by gradually changing the parameters of the Hamiltonian from $x_i^\mu$ to $x_f^\mu$. This is the content of the adiabatic approximation.

Usually, we want to do this in a finite amount of time $T$. To increase our chances of ending in the ground state $\left|\Omega(x_f^\mu)\right\rangle$ we would like to minimize energy fluctuations as much as possible. The question is: Given a fixed time $T$, how should we change the parameters $x^\mu$ to minimize energy fluctuations? The answer is to take the geodesic path $x^\mu(t)$.

For now, let us examine protocols that stay as close as possible to the ground-state manifold. Anandan and Aharonov [40] pointed out that the speed of the evolution of a pure state evolving via the Schrödinger equation is proportional to the uncertainty of its energy

$$ds^2 = \mathrm{Tr}\left(\rho(t + dt) - \rho(t)\right)^2 = 2\Delta H^2 dt^2, \tag{74}$$

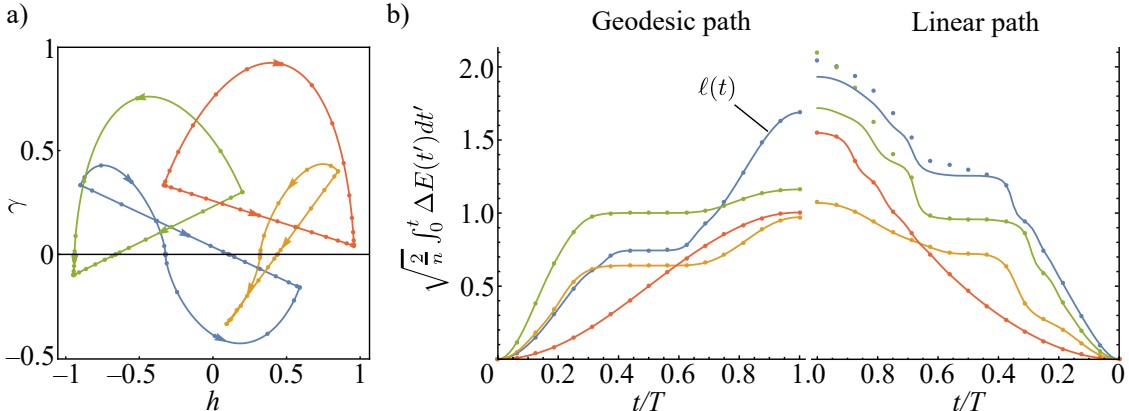

Figure 4: Numerical simulations of different Adiabatic protocols over a finite time $T = 100n$. a) For each color there are two adiabatic protocols: one following a linear path and one following a geodesic path in parameter space. The number of spins in each simulation is $n = 35, 12, 25, 30$ for the blue, orange, green and red curves respectively. Each dot in the path is separated by $T/16$ units of time and the protocol is such that it slows down at the initial, critical and final parameters. b) The points represent the integral over energy fluctuations with respect to time. The solid line is the length of the path $\ell(t)$ given by the Fubini-Study metric. The normalization in the energy variations allows us to compare protocols with different number of spins.

where $\partial_t \rho(t) = i[\rho(t), H(t)]$ and $\Delta H(t)^2 = \text{Tr}(\rho H^2) - \text{Tr}(\rho H)^2$. Or, in other words

$$v = \frac{ds}{dt} = \Delta H. \tag{75}$$

Here $v$ is a velocity. The distance $s(t)$ in this equation is the abstract distance in the projective Hilbert space defined by the Fubini-Study metric. Since the geodesic paths on the ground-state manifold minimize this distance, these are also the paths that minimize the integral ever the energy fluctuations. One might worry that our argument might be too sketchy, but this is indeed the correct answer. A proof of this statement is in [9, 10].

Fig. 4 A shows numerical results supporting this argument. We solved the Schrödinger equation for the evolution of the spin chain of a slowly changing Hamiltonian. To improve the results, we evolved our system using a protocol whose velocity is zero at the initial, critical and final times. We can see this in Fig. 4 a), where points separated by constant time intervals accumulate at the critical line and at the beginning and end of the protocol. We implement this by using functions of the type $A(\exp\{B[1 - \cos^3(t/T)]\} - 1)$. Protocols with this property suppress the fast oscillating terms in the time-evolution that contain the initial excitations of the system [10, 41]. In Fig. 4 b) we see that the integral over energy fluctuations approximates or is larger than the length $\ell(t)$ of the path traced by the time-evolving ground state $|\Omega(t)\rangle$. Since geodesic paths minimize this length, it follows that geodesics are the optimal adiabatic protocols. Note that in some cases the geodesic protocol is well approximated by the linear protocol, like in the case of the yellow trajectories. This is, however, not true for all protocols.

# 8 Conclusion

In summary, we have studied the symmetries of the ground-state manifold of the transverse-field Ising model (TFIM) for both the anisotropic and the isotropic case. Remarkably, some

symmetries in the manifold are not visible at the level of the Hamiltonian. For the anisotropic case, we encountered a hidden symmetry in the ferromagnetic sector of the manifold. This symmetry is related to a change in the magnitude of the magnetic field. The transformation modifies the energy and the states of the system. However, it acts as an isometry on the ferromagnetic sector of the manifold. From this result, we proposed a classification of the different quantum phases of a parameter-dependent Hamiltonian based on the Lie algebra of the related Killing vector fields. We found that the ferromagnetic manifold has two Killing vector fields with an abelian Lie algebra. The paramagnetic manifold has only one Killing vector field and a trivial Lie algebra. We argue that a simple scaling analysis near the critical lines $|h| = 1$ is not enough to determine the Killing vector fields of the metric tensor, since the regular terms in the metric play an important role in defining the isometries of the manifold.

We repeated the analysis for the the Ising limit of the anisotropic TFIM and this resulted in yet more symmetries. The ferromagnetic and the paramagnetic manifold both are maximally symmetric spaces with three Killing vector fields each. The algebra of the ferromagnetic manifold corresponds to the Lie algebra of the Euclidean isometries $\mathfrak{e}(2)$ and is a Type VII$_0$ algebra in the Bianchi classification. The Lie algebra of the paramagnetic manifold is the familiar $\mathfrak{so}(3)$ algebra and is a Type IX algebra in the Bianchi classification.

We took advantage of these symmetries and computed the geodesics of the ground-state manifold for the cases in which enough symmetries were available. Then, we analyzed the behaviour of these solutions near critical lines. We found that some geodesics are confined to specific regions of the ground-state manifold, but there are always solutions that cross the critical lines. These geodesics have several applications in adiabatic quantum preparation protocols as these are the paths that minimize the integral over energy fluctuations.

# Acknowledgements

We would like to thank Anatoli Polkovnikov and Daniel Chernowitz for useful discussions and comments. This work is part of the DeltaITP consortium, a program of the Netherlands Organization for Scientific Research (NWO) funded by the Dutch Ministry of Education, Culture and Science (OCW). Additionally, this work was supported by the Russian Science Foundation Grant No. 20-42-05002.

# A  An expression for the Christoffel symbols

When working with an embedding in a flat manifold, like the set of density matrices in $\mathbb{C}^{n \times n}$, the covariant derivative may be computed by first taking the partial derivative of the vector field and then orthogonally project the result into the tangent space of the embedding

$$\nabla_\mu t_\nu = \Gamma^\lambda_{\mu\nu} t_\lambda = (\partial_\mu t_\nu)^\lambda t_\lambda, \tag{76}$$

note that we are only considering the tangent components of the partial derivative. This result is known as the *Gauss formula*, by applying the dot product with respect to another tangent vector $t_\mu$ on both sides of this equation we get a closed formula for the Christoffel symbols

$$\Gamma^\mu_{\nu\lambda} = g^{\mu\delta} \operatorname{Tr}(t_\delta \partial_\nu t_\lambda), \tag{77}$$

here we take the dot product with respect to the full vector $\partial_\nu t_\lambda$ and not just the tangent projection because the normal components, by definition, vanish. Instead of tangent vectors, we can

express our formula for the Christoffel symbols in terms of bras and kets. Let $\rho = |\psi(x)\rangle\langle\psi(x)|$, then

$$
\begin{aligned}
\Gamma_{\mu\nu\lambda} = \ & (\langle\psi|\partial_\nu\partial_\lambda\psi\rangle - \langle\partial_\nu\partial_\lambda\psi|\psi\rangle)\langle\psi|\partial_\mu\psi\rangle \\
& + (\langle\partial_\nu\psi|\partial_\mu\psi\rangle - \langle\partial_\mu\psi|\partial_\nu\psi\rangle)\langle\psi|\partial_\lambda\psi\rangle \\
& + (\langle\partial_\lambda\psi|\partial_\mu\psi\rangle - \langle\partial_\mu\psi|\partial_\lambda\psi\rangle)\langle\psi|\partial_\nu\psi\rangle \\
& + (\langle\partial_\nu\partial_\lambda\psi|\partial_\mu\psi\rangle + \langle\partial_\mu\psi|\partial_\nu\partial_\lambda\psi\rangle).
\end{aligned}
\tag{78}
$$

A remark: In two dimensions we can compute the inverse metric $g^{\mu\nu}$ easily. So it is feasible to find an expression for the Riemann tensor in terms of the state $|\psi\rangle$ and its derivatives. However, this process becomes tedious, and the resulting expression is long and difficult to handle.

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
