# Peer review of "Hidden Symmetries, the Bianchi Classification and Geodesics of the Quantum Geometric Ground-State Manifolds"

_SciPost Physics, doi:SciPost Phys. 10, 020 (2021)_

## Round 1 · Referee Report · Pieter W. Claeys (Referee 1) · 2020-11-22

Strengths

1- This work explicitly shows how the ground-state geometry of a parameter-dependent Hamiltonian can have symmetries not present in the Hamiltonian itself. An analytical example is given for the anisotropic transverse-field Ising model (TFIM), identifying an unexpected symmetry in the ferromagnetic phase. 2- Explicit construction of Killing vector fields in various regimes of the TFIM. 3- Nontrivial analytical solutions for the geodesics in the ferromagnetic phase of the TFIM. 4- Excellent introduction and discussion of geometric tensors, their properties, and the proposed Bianchi-based classification of different phases. 5- Clear and well-written presentation of both new and known results.

Weaknesses

1- Large overlap with Phys. Rev. B 88, 064304 (2013) (Ref. [8] in the manuscript).
2- Not all of the introduction is relevant for the presented results. 3- Limited discussion of the obtained geodesics.

Report

In this work the authors first present an introduction to the quantum geometric tensor, before focusing on ground-state manifolds. Special attention is paid to isometries and the associated Killing vector fields. The authors then suggest using the Lie algebraic structure of these vectors as a means of obtaining a Bianchi-based classification of quantum ground-state manifolds. The proposed framework is subsequently applied to the anisotropic transverse-field Ising model (TFIM). The geometric tensor is derived, and a symmetry is identified in the ferromagnetic phase that is not present in the paramagnetic phase [Eq. (49)]. The paramagnetic phase exhibits a single Killing vector field/symmetry, whereas the ferromagnetic phase exhibits two. This allows for the analytic calculation of geodesics in the latter, since two conserved charges suffice to fully characterize the geodesics in the considered three-dimensional manifold. The authors additionally characterize the metric near the Ising limit and show how the geodesics can be used to reduce energy fluctuations when varying the underlying Hamiltonian's parameters.

The paper can be divided in two parts, the first being the discussion of ground-state manifolds and geometric tensors in general, and the second being the results for the TFIM. The first part is impressively readable and self-contained and serves as a good introduction to the concepts of the second part. However, I would expect all of the presented concepts to then reappear in the second half, which was not the case. E.g. no mention of the Riemann tensor and Christoffel symbols is made for the TFIM — besides Appendix A claiming that the resulting expressions are long and difficult to handle. Furthermore, it is not immediately clear to me why the authors present all theoretical results for density matrices since only pure states are considered. As such, part of the introduction seems to be disconnected from the authors' results. I do not think this is a major weakness, but it might be useful to further motivate either the introduction of these concepts or why these are not considered in the presented calculations.

The second part presents new results for the TFIM, where the hidden symmetry in the ferromagnetic phase is nontrivial and clearly presented. These results are sound and interesting, and again well-presented in a convincing way. I am especially intrigued by the hidden symmetry from Eq. (49), which the authors rightly claim to be the most important results of the paper. However, a large part of this work derives various results that are already known in the literature: the entirety of Section 4 concerns the calculation of the metric tensor for the TFIM, rederiving the results of Ref. [8], and Section 6.1 visualizes the ground-state manifold using an isometric embedding similar to that presented in Ref. [8]. Could the authors comment on how the visualization from Fig. 3 differs from that in Ref. [8]? Still, the authors generally do make clear which results are known from the literature and which results are new, such that this is also not a major weakness.

My main comment is that it might be useful to extend some of the presented results on geodesics. In Sections 5.2 and 7 the authors derive and apply the geodesics in the ferromagnetic phase. Right now the presented numerics are relatively restricted, and I have two (optional) suggestions here.
First, following Figure 2 it could be interesting to additionally plot the geodesics in the $h\gamma$-plane instead of the $uv$-plane, which might make it easier to understand the geodesic paths in the original coordinates. Extending the number of examples might also help to better understand the structure of the geodesics.
Second, I found the numerical results in Section 7 interesting, but not enough results are presented to be fully convincing. It is clear that the geodesic path leads to a lower energy variance, but the difference only seems to be around a factor two in the lower left panel of Fig. 4— which is insufficient to draw conclusions. Could the authors comment on this? There is some speculation that the energy fluctuations seem to grow linearly rather than exponentially along the geodesic, but I do not think this can be concluded from Fig. 4 alone and would require more extensive numerics. It could be interesting to present numerics for different system sizes to observe the effects there (do the fluctuations scale differently with system size along the two paths?), or present additional numerics for different initial/final Hamiltonians to see if this exponential vs. linear growth is a systematic effect.

To conclude: the results are sound, interesting, and presented in a clear way. I enjoyed reading this paper and am happy to recommend this work for publication in SciPost Physics given that the points raised above are addressed by the authors.

Requested changes

See my report. Additionally, I spotted some minor typos: I suspect Eq. (1) should read $\rho(x)$ instead of $\rho(\lambda)$ and Eq. (16) should have a $\partial_{\nu}$ instead of $\partial_{\mu}$. In Section 6.1 the authors write 'This technique was has been'. In the introduction to Section 7 notation changes from $H(x^{\mu})$ to $H(x_i)$. I also understand the 'isotropic' TFIM/XY model to be the limit $\gamma \to 0$ rather than the limit $\gamma \to 1$ considered in Section 6 titled 'Near the isotropic transverse-field Ising model'. After Eq. (56), the authors write 'where the derivative diverge' .As a minor detail, it would also be useful to make explicit in Eq. (33) which values $k$ can take.

  • validity: top
  • significance: high
  • originality: good
  • clarity: top
  • formatting: perfect
  • grammar: perfect

Author:  Diego Liska  on 2020-12-19  [id 1094]

(in reply to Report 1 by Pieter W. Claeys on 2020-11-22)

Thank you for the precise and detailed report. As a result, we have changed and commented on several points in our paper.

Concerning the introduction, we have moved the piece about Christoffel symbols and curvature tensors to the end of the section. We commented that we do not use these equations explicitly in the rest of the paper. We decided to keep the definitions for completeness, and because we need them to define structures such as Killing vectors and geodesics.

About the overlap of section four, the solution of the XY model, we have commented on a few differences between our paper and the results of Ref. [8]. In our paper, we use the generating functions $S_1$ and $S_2$ (Eqs. 40 and 41), to compute all the components of the metric. We also show how to compute the Berry curvature from $S_3$ (Eq. 42). However, in Ref. [8] the results are presented without any explanation on how to derive them.

We have also changed Figures 2 and 4. Figure 2 now includes more geodesics, and we have depicted them in the $h\gamma$-plane to make them easier to understand. In Figure 4, we decided to improve the numerics. We used a different protocol to ensure that our time-evolved state remains in the ground state. Now, our integral over energy fluctuations matches the length of the path in parameter space. The precise statement says that geodesic protocols minimize this integral, and now the numerics support this statement. We also did this for different paths and degrees of freedom, as you suggested.

Regarding the isometric embedding, the purpose of Figure 2 and Figure 3 is the same: to show the geodesics. The best way to depict geodesics, when $\gamma \rightarrow 1$, is via this embedding.
The figure makes it clear that the somewhat abstract equations we found earlier in the section are the familiar geodesics of the cylinder and the sphere (only now they are combined into the cigar-like manifold).

Finally, we also have corrected the typos and misnames. Again, thank you very much for your suggestions and the detailed analysis of our paper.

Kind regards,

D. Liska
V. Gritsev

---

## Round 1 · Referee Report · Tapobrata Sarkar (Referee 2) · 2020-12-15

Strengths

1) This paper provides a novel direction in research, with an interesting new viewpoint in an existing topic, and has potential for followup works.

2) This is a multi-disciplinary work, involving the information geometry of quantum many body systems, and provides a synthesis of ideas of mathematics in physical systems.

3) The paper is clearly written, and all necessary details have been provided.

Weaknesses

1) A few statements require further elaboration (see report).

2) A few references can possibly be added.

Report

The transverse field Ising model has been studied for a long time. This paper revisits this model from a fairly new perspective. Identifying the Killing vectors from symmetry analysis, the authors give a classification of the ground state for a few cases, using the Lie Algebraic structure of the Killing vectors. This is called the Bianchi classification and the main aim of the authors is to propose such a classification of the ground state of quantum many body systems. I will detail my observations of this paper below.

One of the main outcomes of this paper is the identification of the so called hidden symmetries in the transverse XY model. Namely, a shift in the transverse magnetic field $h$ is found to be a symmetry of the ground state geometry, in the sense that the ground state Riemannian metric does not depend on $h$. This is however not a symmetry of the XY Hamiltonian. The authors mention that eq.(49) of their paper, which quantifies the Killing vector, is their most important result.

I have a few questions here. First, is it "expected'' that the symmetries of the Hamiltonian will naturally carry over to the symmetries of the parameter manifold ? How generic is the authors' result ? Let me make this more precise. The symmetry mentioned above arises from the well known fact that the $h$ coordinate becomes cyclic after a re-parametrisation (see, e.g., the discussion around eqs.(11) and (12) of PRE 90, 042145, or after eq.(24) of arXiv:2005.03532). Such cyclic coordinates are also available for the classical Van der Waals fluid near criticality (see, e.g., eq.(4) of PRE 90, 042145), although the free energy does not possibly possess this symmetry. Why do I therefore "expect" that the information metric will carry precisely the symmetries of the starting model ? I request the authors to comment on this issue.

Secondly, although the Riemannian metric does not depend on $h$, such a symmetry does not exist for the Berry phase. Namely, in the ferromagnetic region, the Berry phase explicitly depends on $h$, see fig.1(a) of PRL 96, 077206. So the full QGT defined in eq.(18) will not have the additional symmetry. This is probably on expected lines. I recommend that the authors add a line to this effect, as it will clarify the issue further.

Finally, can the authors comment on how generic these results are ? The authors state in the introduction that their method gives the possibility of "Bianchi-based classification of the parameter manifolds of the quantum ground states of many-body systems for (at least) a low number of parameters." I point out however that finding a Killing vector might not always be possible even in a sufficiently simple system. If I am not mistaken, the quantum compass model whose geometry appears in arXiv:2005.03532 does not obviously have such vectors. So even in simple quasi-free fermionic models, it is not clear to me if such a classification will be possible. Can the authors please add a comment ?

Next, I come to the issue of geodesics. To my mind, the computation of the geodesics is interesting, but I have a few comments here.

First, in the introduction, the authors state that "... we are not aware of analytical solutions for the geodesic paths of the ground-state manifold of the TFIM spin chain." I think this is not entirely correct. I agree that geodesics on the full parameter manifold, i.e. $(h, \gamma, \phi)$ have not been analytically solved before, but these are known on two-dimensional sections. See, e.g., discussion after eq.(24) of arXiv:2005.03532 or after eq.(12) in PRE 90, 042145.

Secondly, the authors state that "Note that solutions are generically confined to one of the ferromagnetic regions of the ground-state manifold unless there is a fine tuning involved, $Q_{\phi} = 0$ ... ." I think this is also known, see the above mentioned papers, as well as PRE 86, 051117. I think some appropriate references can be given here.

A related query that I have is that from numerical analysis, it is known that the geodesics "turn back" at phase boundaries, unless there is some fine-tuning as the authors mention. A numerical solution of the geodesic equations in the $h-\gamma$ plane appears in fig.(4) of PRE 86, 051117. The authors here on the other hand show that the geodesics touch the boundaries of $h$, fig.(2a). Does this imply that the geodesics "end" there ? If that is the case, it will be a little surprising, as there is no curvature singularity at $h=\pm 1$. I request the authors for a clarification.

Also, the isotropic case has been done on the $h-\phi$ plane, for $\gamma \to 1$. While this is the important Ising limit, and I believe that the results are sufficiently robust for any other finite non-zero value of $\gamma$, I wonder what is the story near the two multi-critical points. Can the authors add some brief comments on this ?

Finally, there is a cosmetic issue. Section 4 computes the Riemannian metric. Since these results are well studied (as the authors say after eq.(46)), I think simply outlining the steps leading to eq.(45) might be enough, and one can probably just give a reference to [34] and/or to PRL 99, 100603, instead of providing the full details, since the community to which this paper will be maximally interesting will already know this. I leave this for the authors to decide.

In summary, I find the paper interesting, the results are novel and add non-trivially to the existing literature, and it has potential for followup work. I recommend publication after the authors address the minor issues raised above.
  • validity: high
  • significance: high
  • originality: high
  • clarity: top
  • formatting: excellent
  • grammar: excellent

Author:  Diego Liska  on 2020-12-19  [id 1093]

(in reply to Report 2 by Tapobrata Sarkar on 2020-12-15)
Category:
answer to question

Dear Dr Sarkar,

We want to thank you for the careful reading of our manuscript and for pointing out several important and relevant references which we unfortunately overlooked. We corrected this in the amended version of our manuscript. Below we answer the questions which were raised in your report.

Best regards

D. Liska
V. Gritsev

Q: I have a few questions here. First, is it "expected'' that the symmetries of the Hamiltonian will naturally carry over to the symmetries of the parameter manifold? How generic is the authors' result? Let me make this more precise. The symmetry mentioned above arises from the well-known fact that the coordinate becomes cyclic after a reparametrization (see, e.g., the discussion around eqs.(11) and (12) of PRE 90, 042145, or after eq.(24) of arXiv:2005.03532). Such cyclic coordinates are also available for the classical Van der Waals uid near criticality (see, e.g., eq.(4) of PRE 90, 042145), although the free energy does not possibly possess this symmetry. Why do I therefore "expect" that the information metric will carry precisely the symmetries of the starting model? I request the authors to comment on this issue.

A: In fact, it is expected because the operator corresponding to the geometric gauge connection $U^{\dagger}\partial_{\alpha}U$ is nothing but a generator of a particular symmetry, corresponding to parameter $\alpha$. One example of this, corresponding to translations, is discussed already in Provost and Vallee paper, but this statement can be easily extended to other generators of any symmetry group of a Hamiltonian.

Q: Secondly, although the Riemannian metric does not depend on h, such a symmetry does not exist for the Berry phase. Namely, in the ferromagnetic region, the Berry phase explicitly depends on , see g.1(a) of PRL 96, 077206. So the full QGT defined in eq.(18) will not have the additional symmetry. This is probably on expected lines. I recommend that the authors add a line to this effect, as it will clarify the issue further.

A: The metric does depend explicitly on the variable $h$. For any metric tensor, we have a well-defined notion of symmetry: the existence of a Killing vector field. If a Killing vector exists, then we can always find coordinates where the metric is independent of a particular variable. However, the notion of Killing vector itself is invariant under coordinate reparametrizations. Maybe we could define similar notions for the Berry curvature using the Lie derivative of the tensor with respect to a particular vector field. Still, the meaning of these symmetries, if they exist, is a topic for another paper.

Q: Finally, can the authors comment on how generic these results are? The authors state in the introduction that their method gives the possibility of "Bianchi-based classification of the parameter manifolds of the quantum ground states of many-body systems for (at least) a low number of parameters." I point out however that finding a Killing vector might not always be possible even in a sufficiently simple system. If I am not mistaken, the quantum compass model whose geometry appears in arXiv:2005.03532 does not obviously have such vectors. So even in simple quasifree fermionic models, it is not clear to me if such a classification will be possible. Can the authors please add a comment?

A: We were not aware of the quantum compass model discussed in arXiv:2005.03532. While the metric presented in eq. 35 of arXiv:2005.03532 looks quite similar to the XY case for $|h| <1$ and fixed $\gamma$, the analysis of its geodesics requires a separate study. We might conjecture that absence of the off-diagonal components of the metric tensor would lead to the presence of some symmetry.

For a general quasi-free fermionic model the information metric of the parameter space is the pull back of the metric from some (possibly high-dimensional homogeneous manifolds associated with classical Lie groups, see arXiv: quant-ph/0606130) so we expect our analysis can be directly applied to this case. For more complicated interacting systems, one should rely on numerical analysis.

Q: Next, I come to the issue of geodesics. To my mind, the computation of the geodesics is interesting, but I have a few comments here. 1) First, in the introduction, the authors state that "... we are not aware of analytical solutions for the geodesic paths of the ground-state manifold of the TFIM spin chain." I think this is not entirely correct. I agree that geodesics on the full parameter manifold, i.e. have not been analytically solved before, but these are known on two dimensional sections. See, e.g., discussion after eq.(24) of arXiv:2005.03532 or after eq.(12)in PRE 90, 042145.

A: Unfortunately, we did overlook these references. We have added them to the manuscript.

Q: Secondly, the authors state that "Note that solutions are generically conned to one of the ferromagnetic regions of the ground-state manifold unless there is a fine tuning involved, ... ." I think this is also known, see the above-mentioned papers, as well as PRE 86, 051117. I think some appropriate references can be given here.

A: Here, we have also added the references to the paper. However, we also clarify an important point. If we restrict ourselves to the $h\gamma$-plane, as it is done in the papers mentioned above, we do expect geodesics to cross the critical line.

As a simple example, note how both curves: $\gamma(\tau) = \tan^2(2 \tau)$, presented in the articles above, and $\gamma(\tau) = $sign$(\tau)\tan^2(2 \tau)$ are solutions to the geodesic equations. Solutions that cross the critical line give the shortest paths between different phases. As long as geodesics touch the critical line $\gamma=0$, we can connect a geodesic from the bottom with a geodesic from the top to get a solution that connects the two phases.

The fine-tuning we discussed in the paper happens when $\phi'(0) \neq 0$. In this case we have that $\gamma(\tau)>\epsilon$ for a positive $\epsilon$. Here, is it not possible to connect with the second phase at $\gamma <0$. We added a paragraph to the manuscript discussing this topic (the paragraph is above eq. 57).

Q: A related query that I have is that from numerical analysis, it is known that the geodesics "turn back" at phase boundaries, unless there is some fine-tuning as the authors mention. A numerical solution of the geodesic equations in the plane appears in g.(4) of PRE 86, 051117. The authors here on the other hand show that the geodesics touch the boundaries of , g.(2a). Does this imply that the geodesics "end" there ? If that is the case, it will be a little surprising, as there is no curvature singularity at . I request the authors for a clarification.

A: We do not agree that with the statement that the curvature does not have singularities at $h=\pm 1$. In fact, some components of the Riemann and Ricci tensors do have this singular behaviour, see Appendix C of arXiv: 1305.0568 for explicit expressions. However, we do expect geodesics to cross the critical line. This fact is clear when we restrict ourselves to the case of constant $\gamma = 1$. In Fig. 3, we pointed out that the geodesics in the cigar-like figure can cross the critical line.

Q: Also, the isotropic case has been done on the $h\phi$-plane, for $\gamma\rightarrow 1$. While this is the important Ising limit, and I believe that the results are sufficiently robust for any other finite non-zero value of $\gamma$

A: The problem when $\gamma$ is not close to one is that the metric is no longer that of a 2-sphere, it has corrections of the order of $(1-\gamma)^2$. We do not expect geodesics to change drastically when this happens. However, we lose one of the Killing vectors in the paramagnetic phase, so the results about their algebra are only valid near this limit.

Q: I wonder what is the story near the two multi-critical points. Can the authors add some brief comments on this?

A: At the multicritical point, the components of the metric diverge. Moreover, their divergence depends on the way we take the limit. Probably, to study the multicritical points, we need to work with a finite-size system, so everything is finite, and then be careful on how to approach the limit.

Q: Finally, there is a cosmetic issue. Section 4 computes the Riemannian metric. Since these results are well studied (as the authors say after eq.(46)), I think simply outlining the steps leading to eq.(45) might be enough, and one can probably just give a reference to [34] and/or to PRL 99, 100603, instead of providing the full details, since the community to which this paper will be maximally interesting will already know this. I leave this for the authors to decide.

A: This issue also overlaps with one of the questions of Dr. P. Claeys (one of the referee of our paper). Our derivation is new: it is based on a generating function, probably not known before, and can be used for other models. On the other hand, the derivations known in the literature are not entirely transparent. For these reasons we would like to keep this part as it is.

---

## Round 2 · Referee Report · Pieter W. Claeys (Referee 1) · 2021-1-5

Strengths

See initial report.

Weaknesses

No major weakness in particular.

Report

All my comments have been addressed, and I am happy to recommend this work for publication in SciPost Physics.

---

## Round 2 · Referee Report · Tapobrata Sarkar (Referee 2) · 2021-1-11

Report

I am okay with the response of the authors and am happy to recommend publication at this stage.

There are just a couple of issues that I raise, although this has little to do with the present version.

1) In response to one of my queries, the authors state that "We do not agree that with the statement that the curvature does not have singularities at $h = \pm 1$. In fact, some components of the Riemann and Ricci tensors do have this singular behaviour, see Appendix C of arXiv: 1305.0568 for explicit expressions."

I would not fully agree with this. Components of the Riemann and Ricci tensors are not scalars, and the divergence of such components is a basis dependent statement, so one cannot really draw too many conclusions from there (meaning that a change of basis might do away with these divergences, although what the new basis means might be another story). What I meant was that the Ricci scalar is non-divergent (but discontinuous nonetheless). This is a coordinate independent statement and in two dimensions is a complete characterisation of the singularity. I am still a little surprised that geodesics can end at $h = \pm 1$ because there is no true (scalar) curvature singularity there. Typically, geodesics end on proper singularities, but maybe this is a matter of finer debate.

2) In a similar spirit, when I said the metric does not depend on $h$, I meant that there exists a coordinate transformation due to which the explicit dependence on $h$ can be done away with. And since one is finally interested in scalars anyway, one metric is as good as another transformed one. In any case, I accept the authors' response here.

In summary, I recommend publication of the manuscript in the present form.

---

## Round 2 · List of Changes

• Added references [28], [29] and [30]
  • Added comments in the introduction
  • Added comments in section 5.2
  • New figures 2 and 4
  • Corrected typos and misnames

---

## Editorial Decision

published